# Vibration Suppression with Use of Input Shaping Control in Machining

**DOI:** 10.3390/s22062186

**Published:** 2022-03-11

**Authors:** Mateusz Kasprowiak, Arkadiusz Parus, Marcin Hoffmann

**Affiliations:** Department of Mechatronics, Faculty of Mechanical Engineering and Mechatronics, West Pomeranian University of Technology in Szczecin, 70-310 Szczecin, Poland; mateusz.kasprowiak@zut.edu.pl (M.K.); arkadiusz.parus@zut.edu.pl (A.P.)

**Keywords:** vibration damping, input shaping control, vibrations during machining process, active tool, zero vibration technique, turbine blade manufacturing, mathematical model of machining process, machining process simulation, extra-insensitive shaper

## Abstract

This article discusses the problem of vibrations during machining. The manufacturing process of generator turbine blades is highly complex. Machining using Computerized Numerical Control (CNC) requires low cutting parameters in order to avoid vibration problems. However, even under these conditions, the surface quality and accuracy of the manufactured objects suffer from high levels of vibrations. Hence, the aim of this research is to counteract this phenomenon. Basic issues related to vibration problems will also be also discussed and a short review of currently available solutions for both active and passive vibration monitoring during machining will be presented. The authors developed a method which does not require any additional equipment other than modified CNC code. The proposed method can be applied to any CNC machine, and is especially suitable for lathes. The method seeks to eradicate the phenomenon of vibrations by providing enhanced control through Input Shaping Control (ISC). For this purpose, the authors present a method for modeling the machining process and design an ISC filter; the model is then implemented in the Matlab and Simulink environment. The last part of the article presents the results, together with a discussion, and includes a brief summary.

## 1. Introduction

Machining is a basic production technique which has been widely used for many years. Nevertheless, contrary to what one might expect, it is a complex process which is accompanied by many unresolved problems. In addition to intentional movements in the cutting process, there are also unintentional movements which often disrupt the proper course of the process. Vibrations are almost always responsible for such movements [1]. Quoting S. Kalinski, vibrations are generally a process, the extent of which changes over time, increasing and decreasing alternately [2]. From the point of view of machining and, above all, taking into account the requirements set by the modern world, this concept is of great importance. Standards regarding the quality of the processed surface, accuracy and dimensional tolerance for the process of cutting, i.e., a basic manufacturing technique, are increasing. The economy of production and productivity is also very important [3]. Therefore, researchers became interested in the problem of vibrations in machining as early as the beginning of the 20th century.

Crucial in the case of vibration damping is the essence of the vibrations, as well as the identification of their type and causes. Vibrations depend not only on the machining tool itself and the technological parameters of the cutting process, but also on the entire machine-tool-work piece system. Due to the fact that this comprises a mass-dispersion-elastic system, it may be assumed that every element, that is, the machine tool itself or its structural elements, the holder, the work piece and the tool [4,5,6,7,8,9], may be the source of the vibrations. A more detailed description of vibration types will be presented in the Section 2.

Over the last century, research on cutting tools, which are largely determined by the aforementioned requirements, has been polarized regarding issues related to vibrations. Over the past few years, dozens of materials and devices for engineering applications with unlimited potential have been invented. The use of intelligent materials in construction and machines offers a lot of possibilities to designers and researchers, mainly due to the ability of these materials to create highly integrated detection, actuation and control functions in engineering systems [5,10,11,12,13,14,15,16,17,18,19,20,21,22,23,24]. Based on the above considerations, according to the authors of this publication, active tools, e.g., those equipped with piezo actuators, are a major niche in the cutting tools market. The proposed solutions are the subjects of research by scientific and research units. Passive tools (e.g., without smart actuators) are also available, but do not give full control over the cutting process. Several vibration suppression methods are described in more detail in the Section 3 [5,11,12,13].

In this paper, particular attention is paid to the so-called smart materials [16], which are used in an increasingly wide range of machining applications. From the point of view of controlling the cutting process, the oscillations that arise during machining cause many undesirable effects, such as difficulties in controlling the object, positioning errors and slower operation of the system, which translates into reduced work efficiency or even the risk of damaging the structure. Very often, basic controllers, such as proportional-integral-derivative (PID) controllers, are insufficient to reduce vibrations in, for example, high inertia systems (cranes or loading cranes, long reach manipulators, etc.) [13,14,15,16,17,18,19,20,21,22].

Nowadays, many ready-made, commercial solutions are available on the market, as well as those remaining only in the laboratory sphere, i.e., controllers and vibration damping systems such as adaptive algorithms using artificial intelligence or advanced digital filters [5,6], mechanical dampers and active damping with feedback and open circuit filters. Mechanical dampers are difficult to design and tune, and generate additional costs. Active methods of damping vibrations with feedback make it possible to achieve very good results, even in the case of nonlinear models. Additionally, they are resistant to modeling errors, but are complicated to implement and expensive.

Therefore, as a matter of principle, machining using with long reach tools presents numerous problems. The high levels of vibrations that occur during such machining are the cause of accelerated tool wear, drastic deterioration of the surface quality or even loss of machining accuracy. Experimental determinations of machining parameters are time-consuming and often do not provide satisfactory results. The use of tools with integrated vibration dampers is costly and not always feasible. Many researchers have attempted to find solutions which can be utilized in the turbine blade manufacturing process. In some cases where vibration problems are caused by the dynamic properties of the machine tools, structural modifications can be introduced in the machine development stage [25]. In [26], problems of manufacturing turbine blades by five-axis milling are analyzed. The authors of that paper note that thin and cantilever shapes cause vibrations, deflections and twists. High levels of vibrations, especially chatter, negatively affect the quality of the machined surface (causing roughness and the waviness in the workpiece). The problem also concerns the turning of long holes, the use of tools with long overhangs or machines which are vulnerable to vibrations. Dynamic issues play an important role in the regeneration of turbine blades [13,14,15,16,17,18,19,20,21,22,23,24,25,26,27,28,29,30,31,32,33,34,35,36,37,38,39,40]. Usually, in this process, overlay welding can be employed to locally apply and solidify the filling materials on a given part. This leads to a local increase of hardness and causes vibration during machining. The methods used to avoid unacceptable vibrations can be roughly categorized into two groups: seeking the best machining parameters and using auxiliary machining equipment. Improving the surface quality of manufactured turbine blades is important for decreasing energy losses in energy production systems [41]. The machining parameters are optimized through the use of genetic algorithms to minimize the surface roughness. The authors of [42] presented a method for continuous spindle speed control with a vibration prediction algorithm intended to improve the quality of machined long thin blades. Counteracting vibrations can be achieved through the use specially designed, dedicated fixtures and clamping devices which fix the turbine blade in place during machining [43,44,45].

An interesting solution to the problem of vibrations during machining may be the use of input shaping control (ISC) [25,26,27,28,29,30,31,32,33,34,35,36,37]. ISC is a very simple algorithm that, by modifying the input signal, is able to reduce the vibrations induced in the system. After changing the forcing signal, vibrations of a specific frequency, depending on the parameters of the system (mainly mass-dispersion-damping parameters), are generated. The ISC method is described in detail in Section 4 as a method to reduce vibrations in a slender cutting tool during turning. The proposed method could also be used with milling machines.

The main aim of the present research is vibration suppression during milling and turning operations in difficult operating conditions, for example, milling turbine blades, which are thin and susceptible to vibrations, as well as turning operations of long holes, using slim and long reach tools. One of the assumptions of this research is the use of noninvasive methods (i.e., which do not interfere in the control system or the construction of the machine) to address vibration issues. Section 4, Section 5 and Section 6 contain an expanded description of our innovative approach, which is presented in the form of a simulation of a turning process (including the modeling machining process as well as the control system design).

## 2. Vibration Types Occurring during Machining

Due to their origin, vibrations occurring during machining can be classified as free, self-excited and forced. Free vibrations are not so harmful; they are caused by a temporary disturbance in the equilibrium state of the system, for instance, through changing movement conditions (braking, starting, load change, entering the material, exiting the material) and are then damped; the degree of energy dissipation depends on the mass-dispersion-elastic properties of the machine tool [3,5,6]. In contrast, self-excited vibrations are a very undesirable, dangerous and harmful phenomenon from the point of view of machining [7]. They are dangerous because as they are created without the involvement of external forces and without changing the parameters of the system; these are vibrations that do not disappear, and their amplitude is self-increasing, resulting in a significant deterioration of the machined surface, an increase in the level of noise emitted, and, in extreme cases, the destruction of the tool. Self-excited vibrations, also called chatter type vibrations, were described by R.N. Arnold in 1946 [9,10]. The graphic below shows the effects of chatter type vibrations during lathing (Figure 1). The figure shows why it is significant to avoid chatter-type vibrations and what they can lead to. Boring operations involving long shafts and the use of cutting tools with a long reach (i.e., more than 10 times the diameter) are most prone to chatter vibrations. Additionally, the machining of slender workpieces may give rise to vibrations under the influence of a low cutting force.

## 3. Preventing Vibrations—A Review

The methods of damping vibrations can be classified according to the type of damper used. We can distinguish passive, semi-active and active dampers. Passive methods are basically characterized by the lack of the ability to generate additional force or the displacement introduced into a machine-tool-work piece system; they are not equipped with a control system and cannot change the operating parameters depending on the machining conditions, which means that the effectiveness of such solutions may be limited.

### 3.1. Passive Solutions

One of the solutions proposed by a leading manufacturer of cutting tools is a tool with an in-built vibration damper. It is an ideal example of passive vibration control. In simple terms, the internal structure of the tool holder can be reduced to a very precisely selected (for undesirable vibration frequencies) mass suspended on two rubber rings and immersed in an oily liquid that increases the damping properties [7,11,12]. A characteristic feature of a this tool is its very strong damping which was verified by the authors via an impulse test using a modal hammer.

From the information that can be obtained from the manufacturer, the tool provides very favorable results. As an example, the rough machining of a process connection flange using a boring cutter with the vibration dampening module is given. The use of a built in passive damper tool holder allowed us to obtain twice the spindle speed, which shortened the unit time by one third and increased productivity by 188%. The tool can also be used for short reach tasks to increase the surface quality and productivity [11].

However, the parameters of the damping module are not modifiable (the mass of the damping module was selected in such a way as to dampen certain vibration frequencies). For other frequencies, the efficiency of the tool may significantly decrease. Hence, semi-active or active vibration control is a better solution. In this respect, a properly selected control system is key. As indicated in the literature, basic control systems such as proportional–integral–derivative (PID) controllers or fuzzy PID controllers [13,14] are used, as well as artificial neural networks and optimal control systems, such as Linear-quadratic-Gaussian (LQG) controllers based on a linear-quadratic algorithm, the aim of which is to minimize the mean square error, or even more advanced adaptive controllers based on the Least mean squares (LMS) algorithm [15].

The executive element, that is, the so called actuator, is also significant from the point of view of the effectiveness of such systems. The most frequently used actuators are piezoelectric transducers, mainly due to their numerous advantages, i.e., low weight, small dimensions, precise movements and ability to work in a wide frequency range [16].

### 3.2. Active Vibration Control—AVC Module

The problem of vibrations concerns practically all machining methods [17]. The authors of this publication propose an innovative, mechatronic piezoelectric module for monitoring vibrations during milling. The innovation of the module is mainly based on the fact that during milling, the damper is able to self-assess and adapt to changing machining conditions [17]. The solution proposed by the authors comprises a light and compact structure that can easily be integrated into micromachine tools. The device connects the electromechanical spindle to the vertical axis of the tool by means of three piezoelectric actuators which measure the relative movement of the two platforms. The idea of the entire structure is based mainly on the Stewart platform, but it only has three degrees of freedom (two degrees of rotation in the *x* and *y* axes, and a shift in the *z* axis).

Two platforms made of Al alloy are fixed relative to the vertical axis of the machine, while the moving platform is connected to the mandrel frame. Three piezo actuators measure the relative movement of the two platforms. The functional concept focuses on recognizing undesirable displacements at the point of tool operation. When undesirable displacements at a given point are detected, three actuators are activated to dampen vibrations, thereby reducing their impact on the treated surface [17]. Piezoelectric actuators are able to handle compressive axial loads. Unfortunately they are very sensitive to tensile or shear forces. Therefore, special connectors integrated into the Active Vibration Control (AVC) module are used to prevent the destruction of the actuators through, for example, shear forces. These connectors are able to eliminate torsional and shear stresses on piezo actuators. The mechanical system ensures the proper rigidity of the actuators, avoiding unwanted stresses and breaks (two additional springs are located near each drive). They have high torsional and radial stiffness, but are free to move in the axial direction. This innovative solution worked successfully, as confirmed by the results showing the displacement of the tool tip over time via the control system [17].

### 3.3. Active Tool Holder for Optimal Tool Position Control

Another quite interesting approach to avoid tool vibrations is isolation from the machine structure, thereby avoiding vibrations caused by the tool’s interaction with the workpiece. The authors of [18] described the construction of an active tool holder which was capable of isolating the cutting tool from the vibrations of the machine tool structure. Isolation was achieved through a control strategy based on the Kalman estimator, a magnetostrictive actuator with high throughput and two accelerometers. The proposed control technique focused on reducing the force transmitted to the tool that was subject to vibration (base excitation) in the presence of disruptions in the cutting process (shavings, trace regeneration, etc.). Machining experiments showed that the proposed technique resulted in an improvement of 25% in terms of the surface roughness of the work piece.

As mentioned, the damping of vibrations takes place by means of an actuator with a large bandwidth. The accelerometer mounted on the cutting tool inside the tool holder then provides feedback for the control elements. The shape, housing and dimensions of this active tool holder are very similar to those of conventional passive tool holders. In addition to having high mechanical strength, this technology has a geometry that is compatible with most existing modern lathe centers [10,19].

### 3.4. Chatter Control System

According to the authors of [9,20,21], vibrations occurring not in the direction of the cutting depth, but rather, in the direction of the cutting motion, have the greatest impact on the quality of machined surfaces, as shown in Figure 2. The authors also noted that the vibration spectra were correlated not with the speed of the work piece (i.e., the source of the vibrations is not the work piece or the spindle) but with the first vibration form of the natural frequency of the tool holder. This, in turn, prompted researchers to take a completely different approach, namely to stiffen the tool holder for the appropriate modes of vibrations.

In order to eliminate and reduce vibrations, it was decided to actively control the vibrations of the tool holder. This means that the first form of the tool natural frequency can be reduced by “antivibrations” using the secondary vibrations generated by a piezoelectric actuator that can be electronically controlled.

A piezoelectric actuator and a sensor providing information about the tool displacement were embedded in the tool holder, and an adaptive control system that can adapt to changing machining conditions on an ongoing basis was developed. As the authors of the publication stated [20], several control algorithms were tested during their research. However, the results were presented using the x-LMS control algorithm, i.e., the FIR filter-based algorithm for adapting the control system, and the LMS algorithm to minimize the mean square error. The given algorithm turned out to be a good solution because it is fast, stable, accurate and easy to implement [10,19].

The authors conducted cutting tests at the CNC Mazak Quickturn 250 lathe center. Thanks to the active vibration control through the use of an active tool, the fundamental frequency suppression was greater than 40 dB, which means that 99% of the resulting vibrations were suppressed. The Ra roughness decreased from 9 µm to 1 µm, while the Rmax decreased from 38 µm to 8 µm.

## 4. Control through Input Shaping Control (ISC)

The problem of undesirable vibrations affects to many fields of technology and concerns both mechanical systems, where there are, for example, elastic, spring or damping connections, as well as electronic systems, where inductive and capacitive elements can cause oscillations. For years, more and more emphasis has been placed on improving the efficiency of devices, especially industrial ones. Weight is also important, and typically, reducing weight has a number of negative consequences, with the problem of oscillations only increasing due to the overly flexible construction of a device or an increase in the speed of the working movements. Taking these aspects into account, vibration control becomes a very important issue [23].

Control systems of devices susceptible to vibrations are a huge area of research. As noted in the previous section, much of the work in this respect has focused mainly on control systems with feedback loops. Unfortunately, this requires the use of additional measuring sensors, a complex control system and higher computational cost.

A compromise solution is so-called impulse control, the great advantage of which is the simplicity of its design and its open feedback loop operation. ISC is a very simple algorithm that, by modifying the input signal, is able to reduce the vibrations induced in the system. After changing the forcing signal (Figure 3), vibrations of a specific frequency, depending on the parameters of the system (mainly mass-dispersion-damping parameters) are generated. For systems based on impulse control, the forcing signal is changed in such a way that the set value signal supplied to the input of the object does not induce vibrations at this specific frequency [24]. As mentioned, the algorithm has many advantages; unfortunately, its control is prone to disturbances and errors related to the model [25,39].

ISC is easy to implement, and its control system makes it possible to work in real time. The vibrations induced by the first part of the control signal (Figure 3) are compensated for by the vibrations induced by the next part of the control signal [39]. The input filter placed between the control signal and the object is modified by the control signal, as shown in Figure 4. The filter shows high efficiency, especially if the natural frequency is known. Feeding the input impulses in counter-phase to the occurring oscillations must occur in an appropriate sequence, i.e., they must be given with a strictly defined amplitude and within a strictly defined time. However, it often happens that information about the control object is incomplete, the natural frequency is unknown or the model state vector is immeasurable. In such cases, a complicated object identification algorithm can be used, but this has a negative impact on the algorithm’s operating time, and its use is justified only when the parameters of the controlled object change during operation or the object shows a high degree of nonlinearity [25,26,27,39].

In order to design a filter with which to shape the signal, it is necessary to determine a sequence of impulses such that the impulse response of the object to the next impulse eliminates the vibrations caused by the previous impulse. As a result, with well-chosen coefficients, the vibrations of the system are damped after the last impulse has occurred [27,39]. The principle of this procedure is illustrated in Figure 4.

Function h(t) describing the designed n-impulse filter may be determined as follows:(1)$$h\left(t\right)=IS\left(t\right)=\underset{i=1\ldots n}{\sum }A_{i}\delta \left(t-t_{i}\right),\;C\leq t_{i}<t_{i+1},A_{i}\neq 0$$
where 

$A_{i}\;\mathrm{is}\;\mathrm{the}\;\mathrm{amplitude}\;\left(i\right)\;\mathrm{of}\;\mathrm{this}\;\mathrm{impulse},\;\mathrm{and}\;\delta \;\mathrm{is}\;\mathrm{the}\;\mathrm{Dirac}\;\mathrm{impulse},\;\mathrm{shifted}\;\mathrm{in}\;\mathrm{time}\;\mathrm{by}\;t_{i}$.

From the above Equation (1), it can be seen that the IS filter has the form of the sum of the sequence of delayed input signals with $A_{i}$ weight and $t_{i}$ delay. In order to obtain the response of the filter in the time domain, the filter should be convolved with any input signal, as described in the Equation (2) [25,39]:(2)$$v\left(t\right)=h\left(t\right)*f\left(t\right)=\int_{-\infty }^{+\infty }h\left(\tau \right)f\left(t-\tau \right)d\tau =\int_{-\infty }^{+\infty }\left(\sum A_{i}\delta \left(t-t_{i}\right)\right)f\left(t-\tau \right)d\tau =\sum A_{i}f\left(t-t_{i}\right),$$
where



v(t) is the modified input signal with an impulse sequence



h(t) is the IS filter



f(t) is the primary forcing signal



As shown in Figure 4, the primary forcing signal is intertwined with the IS filter and then fed to the input of the object G. The filter can be described by the following equation:(3)$$IS\left(t\right)=A_{1}\delta \left(t-t_{1}\right)+A_{2}\delta \left(t-t_{2}\right),$$
(4)$$\left[\begin{array}{c} A_{i}\\ t_{i} \end{array}\right]=\left[\begin{array}{ccc} \frac{1}{1+2K+K^{2}} & \frac{2K}{1+2K+K^{2}} & \frac{K^{2}}{1+2K+K^{2}}\\ 0 & 0.5T_{d} & T_{d} \end{array}\right],$$
(5)$$K=e^{\left(\frac{-\vartheta \pi }{1-\vartheta^{2}}\right)},{\;\omega }_{d}=\omega_{n}\sqrt{1-{Ϛ}^{2}},\;\;\sum A_{i}=1$$
where



$T_{d}\;\mathrm{is}\;\mathrm{the}\;\mathrm{period}\;\mathrm{of}\;\mathrm{damped}\;\mathrm{oscillations}$



ϑ is the logarithmic decrement of damping



In principle, many techniques for shaping input signal impulses have been devised, from one of the simplest ZV algorithms (Eng. Zero Vibration) to CSVS (Component Synthesis Vibration Suppression). For the sake of clarity, all the ISC filters tested by the authors are presented below and are briefly characterized [25,39].

### 4.1. ZV

This input impulse shaping algorithm is the simplest; it was developed in the 1950s [28]. The only limitations for a given algorithm are the minimum time and zero residual vibration. Unfortunately, the algorithm is very sensitive to modeling errors; therefore, the model parameters must be very carefully selected for the algorithm to be effective. The filter is designed in the time domain (instruction), and the effect of the ZV algorithm is a time coarse in the form of impulses with amplitude $A_{i}$ and time $t_{i}$ (see Table 1) [27].

### 4.2. ZVD

ZVD, in contrast to ZV, is an algorithm developed by integration, which makes it more resistant to modeling errors, but at the cost of extending the system response time to the set value. Additional condition $\frac{\partial V\left(\xi \omega \right)}{\partial \omega }=0\;$ means that function V reaches the extreme for the selected pulsation, which, in turn, means that in the vicinity of this pulsation, the amplitude of the residual vibrations is small. Finally, an input signal consisting of three impulses is obtained (Table 1) [27,28].

### 4.3. ZVDD

Another development of the ZV algorithm is the ZVDD filter, which takes into account two natural frequencies dedicated to second-order objects. The principle of the filter operation is the same those of the ZV and ZVD algorithms. As a consequence, an input signal consisting of four impulses (Table 1), determined on the basis of the equations listed in Table 1, is obtained [27,28].

### 4.4. ZVDDD

The ZVDDD algorithm, similarly to the ZVD and ZVDD algorithm, is supplemented by an additional condition, i.e., $\frac{\partial V\left(\xi \omega \right)}{\partial \omega }=0\;$, and takes into account three natural frequencies (see Table 1), which makes the spectrum of operation wider in terms of the frequency of a given filter. Unfortunately, this negatively affects the shaping time, i.e., each additional condition resulting from the derivative of the function V extends the algorithm’s operation by half the natural frequency period [37].

### 4.5. UMZV Algorithm

Compared to the basic ZV filter, the UMZV algorithm is much faster, but has the disadvantage of lower resistance to modeling errors. During one-third of the oscillation period, the UMZV filter has a shorter lifetime than the standard ZV algorithm. Moreover, the shaping of the signal can be compared to the ON-OFF signal (Table 1) which controls, e.g., the actuator [29,30].

### 4.6. EI

The EI (Extra Intensive) algorithm shows high efficiency in the case of controlling models with many forms of vibration. The algorithm is slightly different in comparison to the ZV, ZVD, ZVDD filters (Table 1), because it does not aim at complete oscillation damping in the frequency range designed for a given filter. The assumption is that oscillations are minimized to a given acceptable level of residual vibration. The time of impulse occurrence is the same as in the case of the algorithm, but with a different impulse amplitude, which makes the EI filter much less sensitive to modeling errors and disturbances. The EI filter is applied to systems in which slight vibrations are allowed. The parameters of the model may change significantly [30,31,32,33].

### 4.7. SNA

SNA is a filter designed with a view to balance between speed and resistance to modeling errors and disturbances. The filter is characterized by the course of impulses in the form of positive–negative–positive (Table 1). The filter coefficients are determined primarily by the negative impulse marked in the equations as “b”, on the basis of which the remaining, positive impulses and times are determined; the greater the value of negative impulses, the shorter the shaping time [34,35].

### 4.8. MIS

Theoretical analysis shows that MIS (Modified Input-Shaping) has better performance compared to traditional techniques of input signal shaping [10]. The modified ISC technique does not impose restrictions on the use of the minimum number of impulses (Table 1). By using the MISZV filter with an increased number of impulses, it is possible to achieve zero residual vibration for the modeled frequency, but with the disadvantage of longer filter operation time compared to the ZV filter. The modified input shaping (MIS) technique simplifies the selection of the appropriate filter coefficients, eliminating the need to use numerical optimization, which is necessary in the case of some of the above-mentioned ISC techniques [29,32,36].

Table 1 presents the shapes of the input signals along with the equations to determine the time and amplitude of individual pulses, where ς is the damping coefficient, $\omega_{n}$ is the natural frequency, and $\omega_{d}$ is the damped natural frequency.

## 5. Modeling of the Machining Process with Implementation of ISC

Impulse control is most often used with objects which are susceptible to the formation of vibrations, e.g., port cranes or industrial manipulators [7,8]. The authors propose the shaping of input impulses in the cutting process, specifically, for machining long workpieces which are susceptible to vibrations or cutting with long reach tools. In the authors’ opinion, this approach is innovative. The control technique, based on shaping the input impulse, may have great potential in terms of its ability to dampen vibrations during machining. This will be verified later in the article on the basis of a simulation model.

### 5.1. Mathematical Model

In order to verify the effectiveness of the impulse control to dampen vibrations during machining, a mathematical model of the cutting process is necessary. For this purpose, the authors present below a proposal of a lathe model based on a set of differential equations on the basis of which the state space model was built [7,8,38]. The model of the tested object consists of three basic systems: the cutting tool (lathe knife), the cutting force and the work piece (Figure 5).

Figure 5 shows a graphic interpretation of the mathematical model below, where $m_{1}$ is the mass of the tool, $m_{2}$ is the mass of the work piece, $k_{1}$ is the tool stiffness parameter, $k_{2}$ is the work piece stiffness parameter, $c_{1}$ is the coefficient determining the damping properties of the tool, $c_{2}$ is the coefficient determining the damping properties of the work piece (the value of the coefficient is defined in Table 2), and $y_{1}$ and $y_{2}$ are the movement of the work piece and the cutting tool, respectively.

The system is a typical mass-dispersion-elastic model, where the cutting force acts on two individual masses (i.e., those of the work piece and the tool), taking the opposite direction depending on the direction of operation.


(6)
$$k_{1}y_{1}\left(t\right)+c_{1}{\dot{y}}_{1}\left(t\right)+m_{1}{\ddot{y}}_{1}\left(t\right)=-F_{p}\left(t\right),$$



(7)
$$k_{2}y_{2}\left(t\right)+c_{2}{\dot{y}}_{2}\left(t\right)+m_{2}{\ddot{y}}_{2}\left(t\right)=F_{p}\left(t\right),\;$$


Object state vector:(8)$$\left[\begin{array}{c} x_{1}\left(t\right)\\ x_{2}\left(t\right)\\ x_{3}\left(t\right)\\ x_{4}\left(t\right) \end{array}\right]=\left[\begin{array}{c} y_{1}\left(t\right)\\ {\dot{y}}_{1}\left(t\right)\\ y_{2}\left(t\right)\\ {\dot{y}}_{2}\left(t\right) \end{array}\right],$$
where $x_{1}\left(t\right)$ is tool displacement, $x_{2}\left(t\right)$ is tool velocity, $x_{3}\left(t\right)$ is work piece displacement, and $x_{4}\left(t\right)$ is work piece velocity.
(9)$$\{\begin{array}{c} \frac{dx_{1}}{dt}=x_{2}\left(t\right)\\ \frac{dx_{2}}{dt}=-\frac{k_{1}}{m_{1}}x_{1}\left(t\right)-\frac{c_{1}}{m_{1}}x_{2}\left(t\right)-\frac{F_{cut}\left(t\right)}{m_{1}}\\ \frac{dx_{3}}{dt}=x_{4}\left(t\right)\\ \frac{dx_{4}}{dt}=-\frac{k_{2}}{m_{2}}x_{3}\left(t\right)-\frac{c_{2}}{m_{2}}x_{4}\left(t\right)+\frac{F_{cut}\left(t\right)}{m_{2}}-\frac{F_{p}\left(t\right)}{m_{2}} \end{array},$$

In order to facilitate the implementation of the model and the synthesis of the control system, the model may be written in the form of state space equations:(10)$$\dot{x}\left(t\right)=Ax\left(t\right)+Bu\left(t\right),$$
(11)y(t)=Cx(t)+Du(t),
where
(12)$$A=\left[\begin{array}{cccc} 0 & 1 & 0 & 0\\ -\frac{k_{1}}{m_{1}} & -\frac{c_{1}}{m_{1}} & 0 & 0\\ 0 & 0 & 0 & 1\\ 0 & 0 & -\frac{k_{2}}{m_{2}} & -\frac{c_{2}}{m_{2}} \end{array}\right],$$
(13)$$B=\left[\begin{array}{cc} 0 & 0\\ 0 & -\frac{1}{m_{1}}\\ 0 & 0\\ \frac{1}{m_{2}} & 0 \end{array}\right],$$
(14)$$C=\left[\begin{array}{cccc} 1 & 0 & 0 & 0\\ 0 & 1 & 0 & 0\\ 0 & 0 & 1 & 0\\ 0 & 0 & 0 & 1 \end{array}\right],$$
(15)D=0,
(16)$$\dot{x}\left(t\right)=\left[\begin{array}{cccc} 0 & 1 & 0 & 0\\ -\frac{k_{1}}{m_{1}} & -\frac{c_{1}}{m_{1}} & 0 & 0\\ 0 & 0 & 0 & 1\\ 0 & 0 & -\frac{k_{2}}{m_{2}} & -\frac{c_{2}}{m_{2}} \end{array}\right]\times \left[\begin{array}{c} \begin{array}{c} x_{1}\left(t\right)\\ x_{2}\left(t\right) \end{array}\\ x_{3}\left(t\right)\\ x_{4}\left(t\right) \end{array}\right]+\left[\begin{array}{cc} 0 & 0\\ 0 & -\frac{1}{m_{1}}\\ 0 & 0\\ \frac{1}{m_{2}} & 0 \end{array}\right]\times \left[F_{cut}\right],$$
(17)$$y\left(t\right)=\left[\begin{array}{cccc} 1 & 0 & 0 & 0\\ 0 & 1 & 0 & 0\\ 0 & 0 & 1 & 0\\ 0 & 0 & 0 & 1 \end{array}\right]\times \left[\begin{array}{c} \begin{array}{c} x_{1}\left(t\right)\\ x_{2}\left(t\right) \end{array}\\ x_{3}\left(t\right)\\ x_{4}\left(t\right) \end{array}\right]+0\times \left[F_{cut}\right],$$
(18)$$u\left(t\right)=\left[F_{cut}\right],$$
where $F_{cut}\left(t\right)$ is the cutting force

With the tool model declared in this way, it is now necessary to model the last of the missing elements, i.e., the cutting force model. The cutting force model is described by Equation (19).
(19)$$F_{cut}\left(t\right)=\left(\left(f_{z}-\left(x_{1}\left(t\right)-x_{2}\left(t\right)\right)\right)a_{p}\right)K,$$
where $f_{z}\;\left[\mathrm{mm}\right]$ is the feed value, $a_{p}\;\left[\mathrm{mm}\right]$ is the width of the cutting layer, and $K\;\left[\frac{N}{{\mathrm{mm}}^{2}}\right]$ is the specific cutting resistance coefficient.

The cutting force depends on the feed, the width of the cut layer, and the displacement of the tool and the work piece. In turn, *K* is the specific cutting resistance coefficient, which is determined experimentally and depends on such parameters as tool type, rake angle, clearance angle, type of processed material and machining parameters, among others.

In the model presented above, a certain simplification was made, namely, the bending force was not taken into account. As such, the model assumes the operation of force on one axis only.

### 5.2. Simulation Model

Based on the above-described mathematical model, the authors performed a simulation model in which Equations (1)–(20) were implemented in the Matlab Simulink environment (Figure 6). The above diagram of the simulation model consists of the model of the breakage space of the cutting process—“Model”, the cutting force model—“Cutting force”, the so-called trace regeneration—"external modulation” and impulse control—“ISC”.

## 6. ISC Filter Design

In order to determine the ISC filter coefficients, first, the damping coefficients and the natural frequency of the system should be determined. The coefficients will be determined on the basis of the recorded response of the system (Figure 7), which will make it possible to determine the maximum values of amplitudes for each period of the harmonic waveform.
(20)$$T=\frac{t_{n}-t_{0}}{n},$$
(21)$$\delta =\frac{1}{n}\log\left(\frac{x\left(t_{0}\right)}{x\left(t_{n}\right)}\right),$$
(22)$$\varsigma =\frac{\delta }{\sqrt{4\pi^{2}+\delta^{2}}},$$
(23)$$\omega_{n}=\frac{\sqrt{4\pi^{2}+\delta^{2}}}{T},$$
(24)$$\omega_{d}=\omega_{n}\sqrt{1-\zeta^{2}},$$
(25)$$T=\frac{2\pi }{\omega_{d}},$$

Subsequently, on the basis of Equations (20)–(25), the logarithmic decrement of damping and the damping dynamics curve are determined which, in turn, make it possible to determine the following:

-

ς=0.0039

-

$\omega_{n}=3.0393\times 10^{3}\;\left[\mathrm{Hz}\right]$

-

$\omega_{d}=3.0393\times 1010^{3}\;\left[\mathrm{Hz}\right]$



Based on the technological parameters (Table 3) and parameters, the ISC filter coefficients $A_{n}$ and $t_{n}$ were determined (Table 4) for the following methods: UMZV, ZV, ZVD, ZVDD, MIS, SNA and EI.

Both filter design and confirmation of the effectiveness of the operation (which will be detailed in Section 7) can be confirmed by the FFT frequency [46,47,48,49,50,51,52] and wavelet [46,47,48,49,50,51,52] analyses. However, the authors decided to analyze the results in the time domain; see Figure 8, Figure 9, Figure 10, Figure 11, Figure 12, Figure 13, Figure 14 and Figure 15.

## 7. Results of the Simulation

Figure 10, Figure 11, Figure 12, Figure 13, Figure 14, Figure 15 and Figure 16 show the results of numerical simulations based on the model described in point 5, with the use of impulse control. During the simulation, the authors verified several ISC algorithms, that is, ZV (Zero Vibration), UMZV (Unity Magnitude Zero Vibration Shaper), ZVD (Zero Vibration and Derivative), ZVDD, MIS (Modified Input Shaping), SNA (Specified Negative Amplitude) and EI (Extra-Insensitive Shaper). Figure 8, Figure 9 and Figure 10 show the simulation results, assuming the use of a finishing machine with the following cutting parameters.

Table 4 also summarizes the ISC parameters for the model described in Section 5, using all of the tested impulse control algorithms in case of finishing machining (Table 3), roughing machining (Table 5) and precision machining (Table 6).

In turn, Figure 11, Figure 12 and Figure 13 show the simulation results during rough machining, and Figure 14 and Figure 15 show the results during medium machining.

## 8. Conclusions

The proposed the technique could be implemented in flexible and precision turn-mill machine tools to enable the production of high-quality turbine blades [45]. In this paper, the use of an ISC technique to reduce vibrations in machining is proposed. A model of a steel shaft lathe process was developed using a tool with a large reach (320 mm). In the absence of the proposed technique, the machining process was accompanied by high-amplitude vibrations occurring from the moment of starting the cutting (see Figure 9) which prevented the process from being properly conducted. Figure 8, Figure 9, Figure 10, Figure 11, Figure 12, Figure 13, Figure 14, Figure 15 and Figure 16 show the same processing variant but with the ISC control applied. There was a clear reduction in the amplitude of vibrations, which the cutting process to occur to completion. ISC only requires modification to the CNC work program; there is no need to use any additional elements, tools, handles, etc. The limitations of the proposed method are related to the dynamic properties of the feed drive, which, in the case of high-frequency vibrations, may lead to a reduction in operating efficiency. In order to determine the parameters necessary to modify the CNC program, it is only necessary to perform a simple impulse test. Once determined, these parameters can be used in future with the same machine tool. Changes in the cutting conditions, e.g., of a work piece, tool or machine configuration, require the impulse test to be repeated.

## Figures and Tables

**Figure 1 sensors-22-02186-f001:**
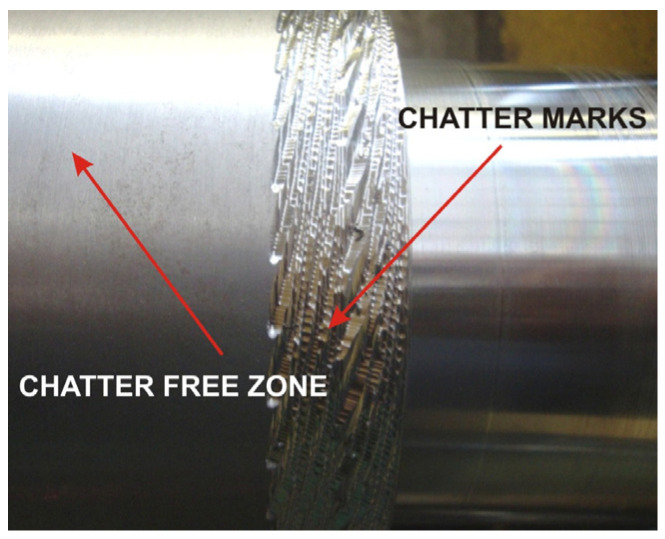
An example of a surface after machining with self-excited vibrations.

**Figure 2 sensors-22-02186-f002:**
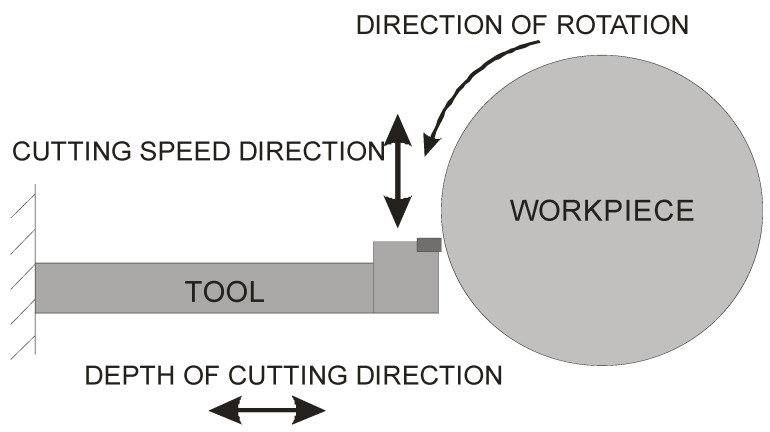
Graphical model of machining by turning.

**Figure 3 sensors-22-02186-f003:**
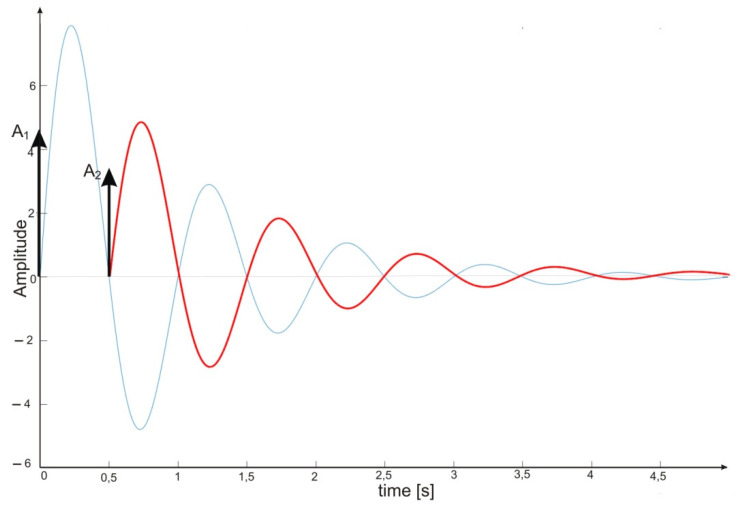
The principle of vibration elimination. The impulse response of the object to impulses A1 and A2.

**Figure 4 sensors-22-02186-f004:**
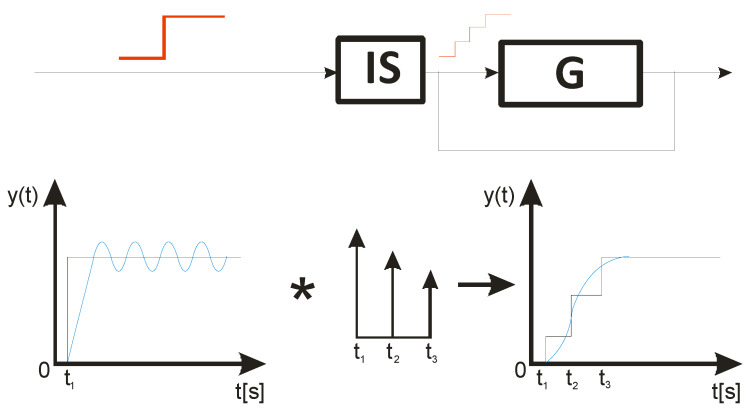
Implementation of a input filter by a convolution operation, where the black waveform represents the input and the blue signal represents the system response.

**Figure 5 sensors-22-02186-f005:**
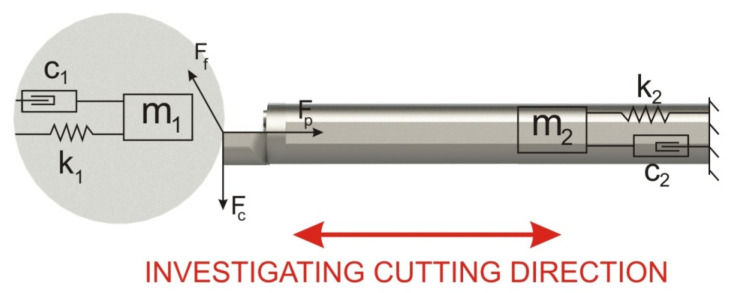
Model of the tool-cutting force-workpiece system.

**Figure 6 sensors-22-02186-f006:**
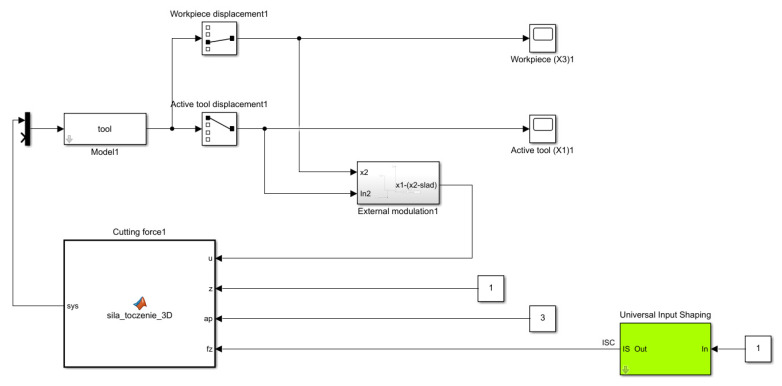
Simulation model in the MATLAB Simulink environment.

**Figure 7 sensors-22-02186-f007:**
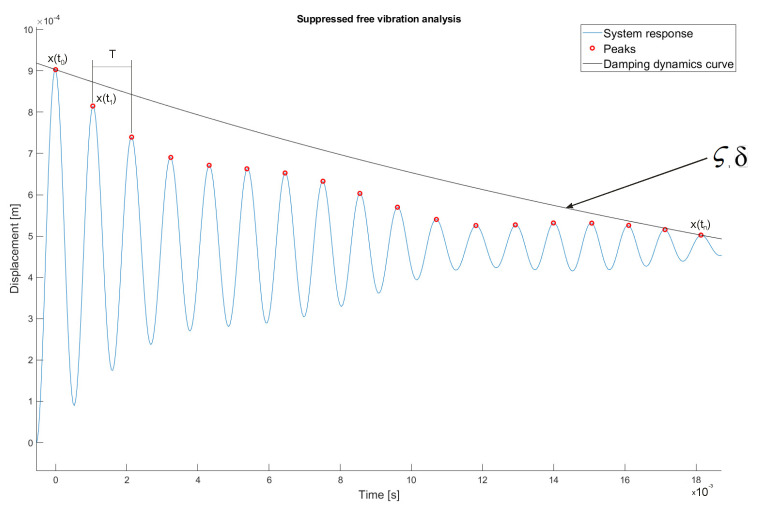
Summary of the recorded response of the system (blue line) with marked peaks of individual amplitudes (red markers) and the damping dynamics curve (black line).

**Figure 8 sensors-22-02186-f008:**
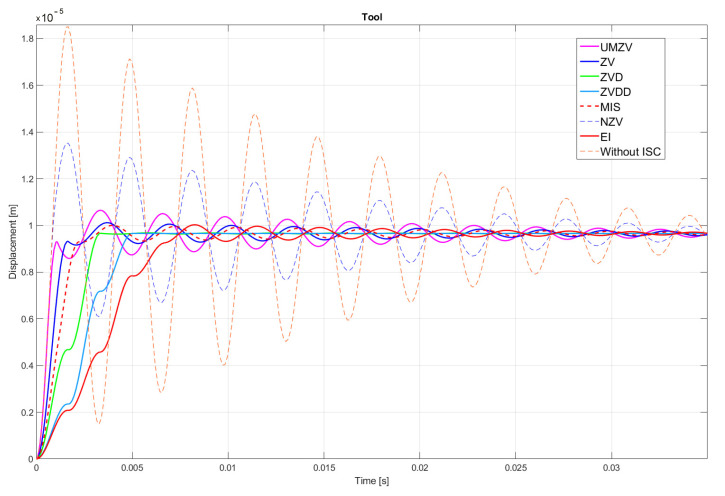
Tool displacement during finishing machining: without ISC control, using UMZV, ZV, ZVD, ZVDD, MIS, SNA and EI algorithm.

**Figure 9 sensors-22-02186-f009:**
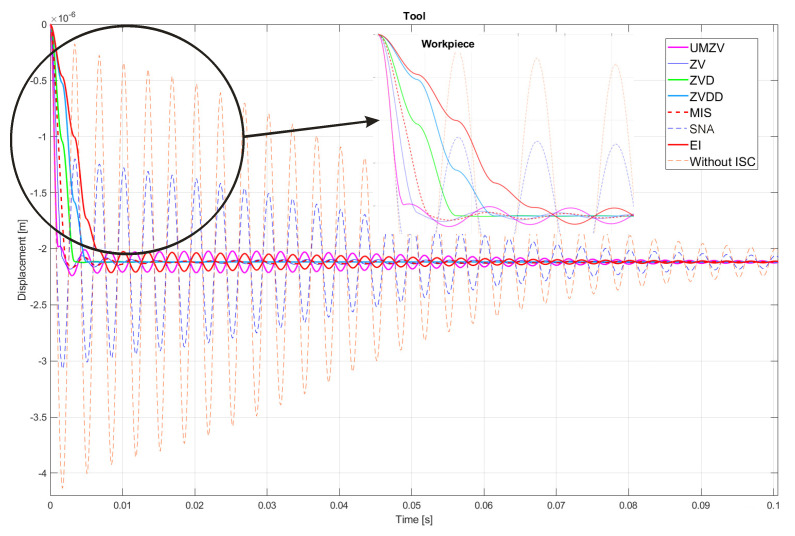
Work piece displacement during finishing machining: without ISC control, using UMZV, ZV, ZVD, ZVDD, MIS, SNA and EI algorithm.

**Figure 10 sensors-22-02186-f010:**
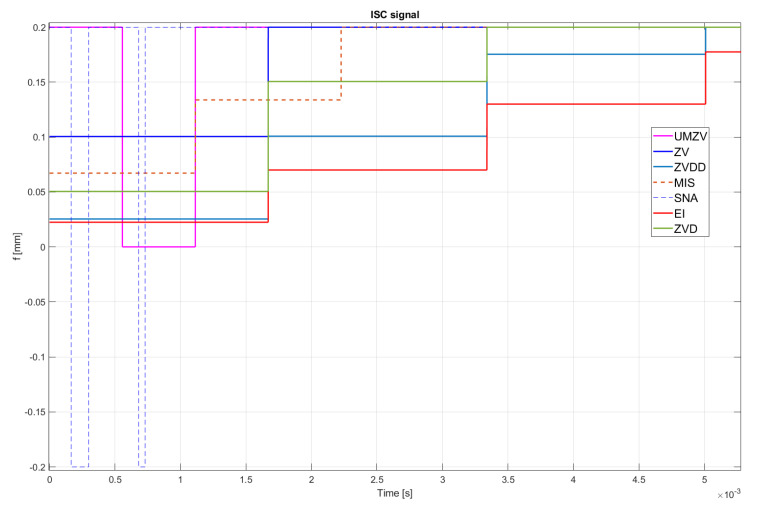
ISC signal of UMZV, ZV, ZVD, ZVDD, MIS, SNA and EI algorithm during finishing machining.

**Figure 11 sensors-22-02186-f011:**
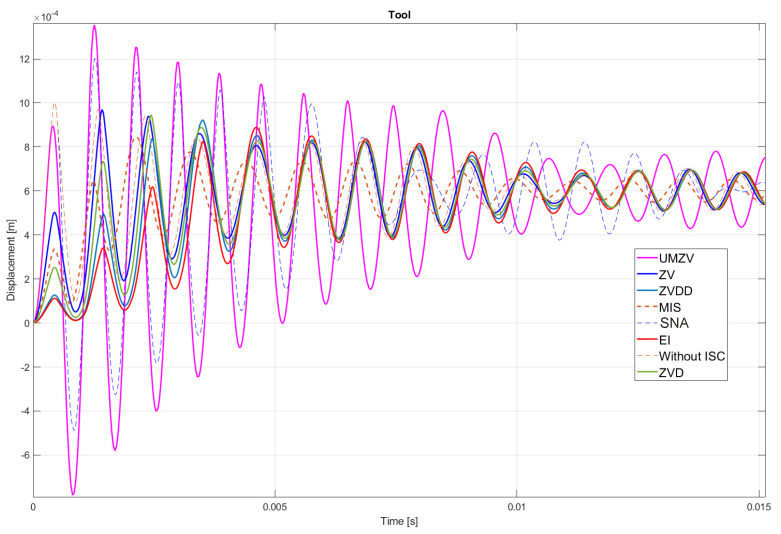
Tool displacement during rough machining: without ISC control, using UMZV, ZV, ZVD, ZVDD, MIS, SNA and EI algorithm.

**Figure 12 sensors-22-02186-f012:**
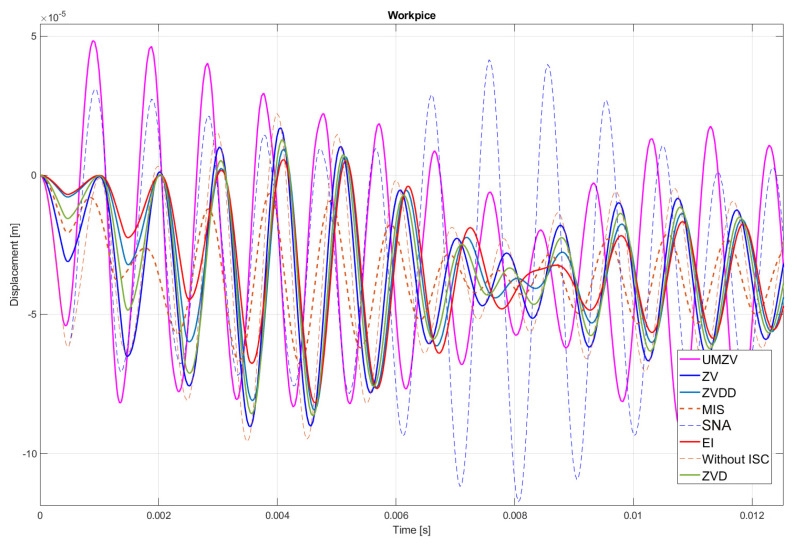
Work piece displacement during rough machining: without ISC control, using UMZV, ZV, ZVD, ZVDD, MIS, SNA and EI algorithm.

**Figure 13 sensors-22-02186-f013:**
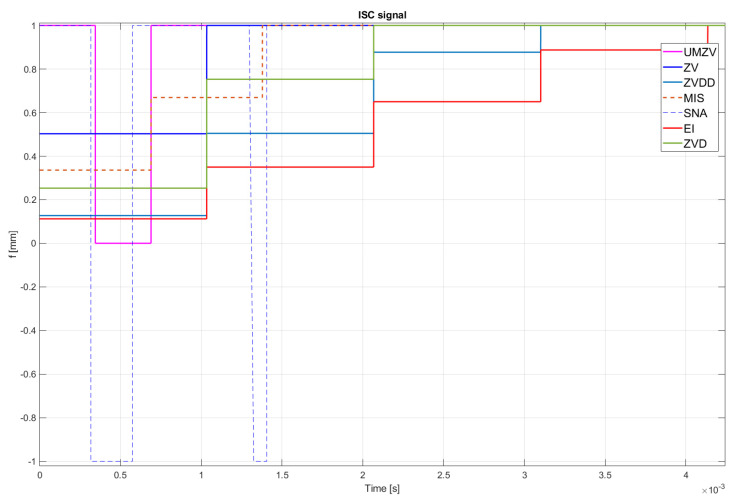
ISC signal of UMZV, ZV, ZVD, ZVDD, MIS, SNA and EI algorithm during rough machining.

**Figure 14 sensors-22-02186-f014:**
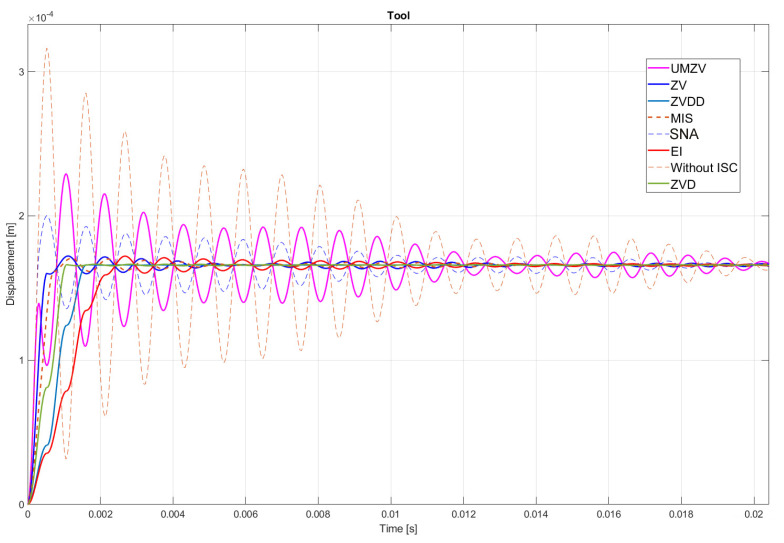
Tool displacement during medium precision machining: without ISC control, using UMZV, ZV, ZVD, ZVDD, MIS, SNA and EI algorithm.

**Figure 15 sensors-22-02186-f015:**
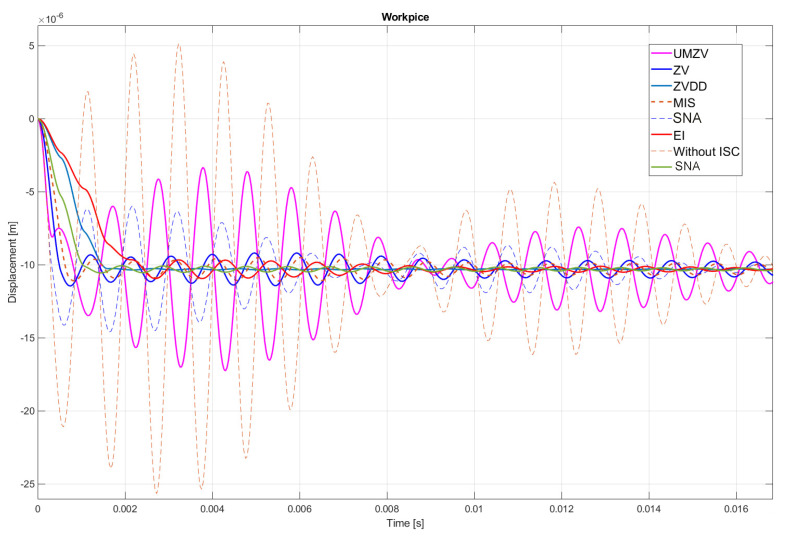
Work piece displacement during medium precision machining: without ISC control, using UMZV, ZV, ZVD, ZVDD, MIS, SNA and EI algorithm.

**Figure 16 sensors-22-02186-f016:**
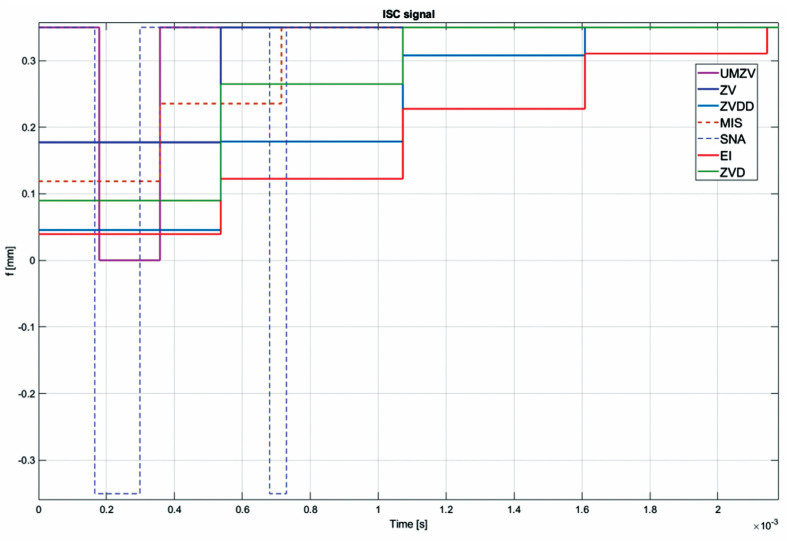
ISC signal of UMZV, ZV, ZVD, ZVDD, MIS, SNA and EI algorithm during precision machining.

**Table 1 sensors-22-02186-t001:** The form of pulses for individual ISC algorithms.

ZV	$\left[\begin{array}{c} A_{i}\\ t_{i} \end{array}\right]=\left[\begin{array}{cc} \frac{1}{1+K} & \frac{K}{1+K}\\ 0 & \frac{1}{2T_{d}} \end{array}\right],\;\;K=e^{\left(\frac{-\vartheta \pi }{1-\vartheta^{2}}\right)}$ [37]
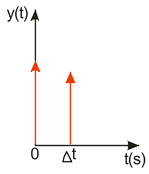	
ZVD	$\left[\begin{array}{c} A_{i}\\ t_{i} \end{array}\right]=\left[\begin{array}{ccc} \frac{1}{1+2K+K^{2}} & \frac{2K}{1+2K+K^{2}} & \frac{K^{2}}{1+2K+K^{2}}\\ 0 & 0.5T & T \end{array}\right]$ [37]
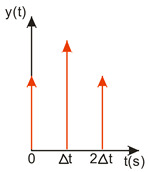	
ZVDD	$t_{1}=0,\;t_{2}=\frac{\pi }{\omega_{d}},t_{3}=\frac{2\pi }{\omega_{d}},\;t_{4}=\frac{3\pi }{\omega_{d}}$ [37] $A_{1}=\frac{1}{1+3K+3K^{2}+K^{3}},$ [37] $A_{2}=\frac{3K}{1+3K+3K^{2}+K^{3}},\;A_{3}=\frac{3K^{2}}{1+3K+3K^{2}+K^{3}},A_{4}=\frac{K^{3}}{1+3K+3K^{2}+K^{3}}$ [37]
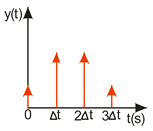	
ZVDDD	$\left[\begin{array}{c} A_{i}\\ t_{i} \end{array}\right]=\left[\begin{array}{ccccc} \frac{1}{D} & \frac{4K}{D} & \frac{6K²}{D} & \frac{4K^{3}}{D} & \frac{K^{4}}{D}\\ 0 & \frac{\tau_{d}}{2} & \tau_{d} & \frac{3}{2}\tau_{d} & 2\tau_{d} \end{array}\right]$ [27,28] $D=1+4K+6K²+4K^{3}+K^{4}$
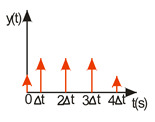	
UMZV	$\left[\begin{array}{ccc} 0 & t_{2} & t_{3}\\ A_{1} & A_{2} & A_{3} \end{array}\right]=\left[\begin{array}{ccc} 0 & \frac{\pi }{3\omega_{n}} & \frac{2\pi }{3\omega_{n}}\\ 1 & -1 & 1 \end{array}\right]$ [29,30] Ϛ=0
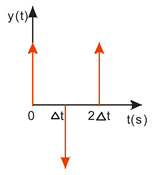	
EI	$\left[\begin{array}{ccc} 0 & t_{2} & t_{3}\\ A_{1} & A_{2} & A_{3} \end{array}\right]=\left[\begin{array}{ccc} 0 & \frac{\pi }{3\omega_{n}} & \frac{2\pi }{3\omega_{n}}\\ \frac{1+V}{4} & \frac{1-V}{2} & \frac{1+V}{4} \end{array}\right]$ [30,31,32,33] $V=e^{-Ϛ\omega t_{n}}\sqrt{A+B}$ [30,31,32,33] where: $A=(\sum_{i=1}^{n}A_{i}e^{Ϛ\omega t_{i}}cos{\left(\omega \sqrt{1-{Ϛ}^{2}t_{i}}\right)}^{2}$ [30,31,32,33] $B=(\sum_{i=1}^{n}A_{i}e^{Ϛ\omega t_{i}}sin{\left(\omega \sqrt{1-{Ϛ}^{2}t_{i}}\right)}^{2}$ [30,31,32,33]
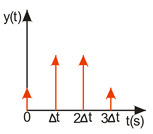	
SNA	$\left[\begin{array}{ccc} 0 & t_{2} & t_{3}\\ A_{1} & A_{2} & A_{3} \end{array}\right]=\left[\begin{array}{ccc} 0 & t_{2} & t_{3}\\ \frac{1-b}{2} & b & \frac{1-b}{2} \end{array}\right]$ [34,35] $t_{2}=\frac{1}{\omega_{n}}cos^{-1}\left(\frac{-b}{1-b}\right)$ [34,35] $t_{3}=\frac{1}{\omega_{n}}cos^{-1}\left(\frac{2b^{2}}{{\left(1-b\right)}^{2}}-1\right)$ [34,35]
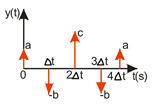	
MIS	$M=K+\ldots +K^{i-1}+K^{n-1}$ [29,32,36] $A_{i}=\left[\begin{array}{ccccc} \frac{1}{1+M} & \frac{K}{1+M} & \ldots & \frac{K^{i-1}}{1+M} & \frac{K^{n-1}}{1+M}\\ 0 & \frac{T_{d}}{n} & \ldots & \frac{\left(i-1\right)T_{d}}{n} & \frac{\left(n-1\right)T_{d}}{n} \end{array}\right]$ [29,32,36] $K=e^{-\frac{2Ϛ\pi }{n\sqrt{1-{Ϛ}^{2}}}}$ [9,32,36]
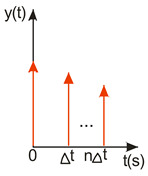	

**Table 2 sensors-22-02186-t002:** Model parameters.

Tool weight	$m_{1}$ = 0.2 kg
Tool stiffness	$k_{1}$$=2.2\;\times \;10^{9}$ N/m
Tool damping	$c_{1}$ = 70 kg/s
Work piece weight	$m_{2}$ = 3 kg
Work piece stiffness	$k_{2}$$=1.7\;\times \;10^{8}$ N/m
Work piece damping	$c_{2}$ = 290 kg/s

**Table 3 sensors-22-02186-t003:** Technological parameters of the cutting process during finishing machining.

Spindle speed	n = 800 rmp
The width of the cutting layer	$a_{p}$ = 0.5 mm
Feedspeed	f = 0.2 mm/r

**Table 4 sensors-22-02186-t004:** Parameters of ISC techniques.

Algorithm	A	t
UMZV	$\left[\begin{array}{ccc} 1 & -1 & 1 \end{array}\right]$	$\left[\begin{array}{ccc} 1 & 0.0006 & 0.0011 \end{array}\right]$
ZV	$\left[\begin{array}{cc} 0.5026 & 0.4974 \end{array}\right]$	$\left[\begin{array}{cc} 0 & 0.0017 \end{array}\right]$
ZVD	$\left[\begin{array}{ccc} 0.2526 & 0.5000 & 0.2474 \end{array}\right]$	$\left[\begin{array}{ccc} 0 & 0.0017 & 0,0033 \end{array}\right]$
ZVDD	$\left[\begin{array}{cccc} 0.1270 & 0.3770 & 0.3730 & 0.1230 \end{array}\right]$	$\left[\begin{array}{cccc} 0 & 0.0017 & 0.0033 & 0.0050 \end{array}\right]$
MIS	$\left[\begin{array}{ccc} 0.3357 & 0.3333 & 0.3310 \end{array}\right]$	$\left[\begin{array}{ccc} 0 & 0.0011 & 0.0022 \end{array}\right]$
SNA	$\left[\begin{array}{ccccc} 1 & -2 & 2 & -2\; & 1 \end{array}\right]$	$\left[\begin{array}{ccccc} 0 & 0.0005 & 0.0009 & 0.0021 & 0.0023 \end{array}\right]$
EI	$\left[\begin{array}{ccccc} 0.1124 & 0.2375 & 0.3002 & 0.2375 & 0.1124 \end{array}\right]$	$\left[\begin{array}{ccccc} 0 & 0.0017 & 0.0033 & 0.0050 & 0.0067 \end{array}\right]$

**Table 5 sensors-22-02186-t005:** Technological parameters of the cutting process—roughing machining.

Spindle speed	n = 800 rmp
Width of the cutting layer	$a_{p}$ = 4 mm
Feed speed	f = 1 mm/r

**Table 6 sensors-22-02186-t006:** Technological parameters of the cutting process—medium-precision machining.

Spindle speed	n = 800 rmp
Width of the cutting layer	$a_{p}$ = 3 mm
Feedspeed	f = 0.35 mm/r

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
