# Peer review of "Vibration Suppression with Use of Input Shaping Control in Machining"

_sensors, 2022, doi:10.3390/s22062186_

Round 1

Reviewer 1 Report

Please see the attached pdf file.

Author Response

Dear Reviewer, 

in the attachment I am sending our reply to your review. 

Kind regards

Mateusz Kasprowiak 

Reviewer 2 Report

Dear Authors,

This work is interesting and important for reducing vibrations in machining by using ISC techniques. The work was done at a high level and is of interest to engineers and researchers. While this work is publishable, there exist a number of issues to be addressed prior to publication.

Specific Comments
1. Lines 52-62, the submitted statements must be supported by references. Please correct.
2. Lines 138-144 require supplementing specific technological applications in which the proposed design of cutting tools is applied.
3. Line 484, the feed unit is incorrectly specified. Please correct.
4. Line 486, please explain how the cutting tool offset was controlled and what this term means.
5. Please describe in more detail the imposition of feedback on the controlled parameter when reducing the offset of the cutting tool.
6. Please describe the experimental setup used in this research paper.
7. Why is the turning scheme presented in the mathematical model, while in the abstract and conclusions, it is indicated that the work investigates the process of multi-axis machining of blades? Please correct.
8. It is recommended in the Introduction to focus more on the technological conditions of milling rather than turning. Since many of the proposed technical solutions in the Introduction for milling are challenging to implement.

Author Response

(The authors gave the same response as above.)

Reviewer 3 Report

The information presented in this article is useful, research contribution is satisfactory. However, the following suggestions are recommended:

  • Your abstract does not highlight the specifics of your research or findings. Rewrite the Abstract section to be more meaningful. I suggest: problems, Aim, Methods, Results, and Conclusion.
  • Nomenclature needs to be provided for clearing the ambiguity of short forms for the benefit of the readers.
  • Introduction suffers from lack of motivations and innovations. Although the results and purpose of the research are interesting, the authors do not clarify what the main objective or scope of the work is in the introduction, so it is necessary to specify it. It should be expanded to include a more detailed discussion of current problems.
  • Authors need to explain in detail what have done and what have not been done in other studies. What is the different within your study and other study?
  • You need to review your English.

Author Response

(The authors gave the same response as above.)

Round 2

Reviewer 1 Report

I think the authors increased the quality of the presentation and tried to address the issues raised in the first round, except for a spectral (FFT-wavelet) analysis. However, this is mentioned as a paragraph and tied to the existing literature. Thus, OK.

There are minor typos in references which can also be corrected in the editorial stage.

For example in 46, a book and a paper are added in a single sentence. Please split those two references, and check all of the list again.

Reviewer 3 Report

It is supported to accept this work. This is a great work.